# Challenges Faced by Health Professionals in Obtaining Correct Medication Information in the Absence of a Shared Digital Medication List

**DOI:** 10.3390/pharmacy9010046

**Published:** 2021-02-22

**Authors:** Unn Sollid Manskow, Truls Tunby Kristiansen

**Affiliations:** Norwegian Centre for E-health Research, University Hospital of North Norway, 9017 Tromsø, Norway; truls.tunby.kristiansen@ehrs.no

**Keywords:** health professionals, digital solutions, medication management, shared medication list, primary health care, patient safety, medication safety

## Abstract

Information about patient medication use is usually registered and stored in different digital systems, making it difficult to share information across health care organisations. The lack of digital systems able to share medication information poses a threat to patient safety and quality of care. We explored the experiences of health professionals with obtaining and exchanging information on patient medication lists in Norwegian primary health care within the context of current digital and non-digital solutions. We used a qualitative research design with semi-structured interviews, including general practitioners (n = 6), pharmacists (n = 3), nurses (n = 17) and medical doctors (n = 6) from six municipalities in Norway. Our findings revealed the following five challenges characterised by being cut off from information on patient medication lists in the current digital and non-digital solutions: ‘fragmentation of information systems’, ‘perceived risk of errors’, ‘excessive time use’, ‘dependency on others’ and ‘uncertainty’. The challenges were particularly related to patient transitions between levels of care. Our study shows an urgent need for digital solutions to ensure seamless, up-to-date information about patient medication lists in order to prevent medication-related problems. Future digital solutions for a shared medication list should address these challenges directly to ensure patient safety and quality of care.

## 1. Introduction

An updated and accurate information about a patient´s current medicine use is important in order to cure and prevent many medical conditions [1]. Medicine-related problems (MRP) such as side effects, inappropriate use and errors are serious threats to patient safety, as they may reduce quality of life, cause morbidity, death and increase health care costs [2,3]. The WHO 3rd Global Patient Safety Challenge: Medication without harm aims to reduce avoidable medication-related harm by 50% globally over the next 5 years [4]. Medication errors have been reported as the third leading cause of death in the US [2] and is linked to substantial financial cost worldwide [5]. At present, information on patient medicine use is registered and stored in different local IT systems that are not able to share data across the health care services [6,7]. Health information exchange (HIE) is an electronic exchange of information across organisations and patient safety is a potential advantage of HIE, provided the right information is available to the right person at the right time [8]. Health professionals typically use both digital and manual sources to obtain information and lack immediate access to an accurate list of patient medications, which increases the risk of medication errors [9]. 

Previous studies show that fragmented medication information causes poor communication and a lack of information flow across services and levels of care, which can lead to potentially harmful medication errors [6,10,11]. Within the primary health care services, studies show the lack of medication information exchange as a reason for medication errors [12,13,14]. Several other national and international studies report problems with accurate information about a patient’s current medication list and how this lack of information can potentially affect medication safety and quality of care [12,15,16,17,18]. A recent narrative review of research on the effects of and experiences with digital solutions for a shared medication list reported a knowledge gap on the effects on patient safety and quality of care [19]. A study from Denmark showed the implementation of a shared medication list in the early phase to be difficult due to technical problems, time consumption, and motivation [20]. The Ministry of Health and Care Services in Norway has acknowledged that medication management within primary health and care services is insufficient because health professionals lack a real-time overview and the possibility for HIE on the patient´s current use of medications across health care facilities and levels of care [21]. 

### The Norwegian Health Care System and National eHealth Solutions

Health care and social services are based on the classic Scandinavian welfare model, combining financing and the provision of universally accessible services, mainly within the public sector [22]. The health care system is roughly organised into two main sectors: (1) primary health care and long-term care and (2) specialised health services (hospitals) [23]. The Norwegian ‘Collaboration in health care reform’ from 2012 increased municipalities’ responsibility to take care of citizens at an earlier stage following discharge from the hospital in order to achieve more coordinated services for patients and meet the growing demand for health services [24]. 

The goal of the Norwegian eHealth strategy is to make the right data available to the right person at the right time, regardless of where the patient has received medical care [25]. In terms of digital medication management, the Norwegian Directorate for eHealth (NDE) is currently developing and implementing the following national solutions: electronic Multidose Drug Dispensing (eMD), Summary Care Record (SCR) to primary health care services (nursing homes and home care services) and a nationwide Shared Medication List (SML) [25]. The goal of the SML is to share updated, structured and complete medication information throughout the entire patient journey across organisations and levels of care, available for health professionals involved in the patient’s treatment and care [25]. The implementation of the eMD and SCR in primary health care represents an important step towards national implementation of the SML. 

This article is part of a pre-study in a larger national mixed-method study during the years 2019–2025 that investigates the effects and experiences on end-users (health care professionals and patients) before, during, and after the implementation of the SML in primary health care. Six municipalities in the process of implementing the SCR and SML are included in the study. The overall objective is to produce research-based knowledge of the introduction of the SCR and SML in the primary health care sector, focusing on access to medication information, HIE, medication safety, efficiency, work processes and interprofessional collaboration. The main study will provide transferrable and decision-relevant knowledge about the impact and conditions of the SML across health care organisations and end-users as a key tool for the health authorities responsible for the implementation. The study consists of three sub-studies: (1) pre-study, (2) early evaluation (during), and (3) after, national implementation of the SML. 

The pre-study consists of a qualitative and quantitative component aimed at obtaining both in-depth knowledge (interviews) and the prevalence of those experiences within a larger population (survey). The objective of the pre-study is to investigate the experiences of health professionals and patients with the medication management process before the implementation of the SML. 

This article presents results from the qualitative part of the pre-study. The aim was to explore how health professionals in primary health care experience access to and the exchange of patient medication lists within the context of the current digital (EHRs, electronic prescribing systems, e-messages, SCR (Appendix A)) and non-digital solutions (manual sources of information, i.e., paper, phone, fax, face-to-face). 

## 2. Materials and Methods

### 2.1. Research Design

We used a qualitative research design based on Grounded Theory [26] to explore the experiences of health professionals with access to and the exchange of patient medicine lists within the context of the current digital and non-digital systems. Grounded Theory argues for an open-minded and unbiased analytical approach [27]. We aimed to approach the material without any fixed set of concepts or expectations and allow patterns to emerge from the material. Semi-structured interviews were conducted in order to acquire the experiences, thoughts and perspectives of the participants, while at the same time making sure that the themes related to the research question were covered [28]. Prior to the study, the interview guide was piloted on two colleagues with clinical experience from primary health care. The interview guide for this pre-study covered the following five themes: access to critical patient information (only for nurses and MDs in nursing homes and home care services due to the upcoming implementation of a SCR), access to medication information, collaboration with other parties, decision support and expectations of the SCR and SML. In this article we present findings related to theme no. 2: Access to medication information (Table 1).

### 2.2. Recruitment, Research Sites and Participants

The project manager (USM) contacted the heads of health and care services in nine different Norwegian municipalities by phone and email, providing them with information about the larger national study and an invitation to participate. The selection of municipalities took place in close communication with the NDE to ensure the inclusion of municipalities based on the following criteria: Had agreed to and/or is in the process of implementing the SCR and SMLA spread of municipalities based on the three main suppliers of EHR systems in Norway (Tieto, Visma and DIPS Front)A geographical spreadSmall, medium and large municipalities.

Seven out of nine municipalities agreed to take part in the study. Two municipalities declined due to a lack of available resources as a result of the implementation of other large IT-systems at that point in time. Of the seven included municipalities, planned interviews for the pre-study were not possible for one due to unexpected circumstances. As a consequence of the Covid-19 pandemic, we did not schedule a revisit to this municipality and decided against including the last two municipalities for the same reason. The authors discussed and concluded that data saturation was achieved with the included 6 municipalities.

To capture the experiences of different health professionals involved in the medication management process in primary health care, we chose to recruit pharmacists, general practitioners (GPs), nurses and medical doctors (MDs) working in primary health care services. As nurses work at different sites within primary health care services, we chose to include nurses from both home care services and different types of nursing homes (long-term, short-term and intermediate departments). In addition, we established the following criteria for participants: representation of both genders, two or more years of experience in the present workplace and experience using the local EHR for medication information. In close communication with the leaders of each municipality, a designated contact person coordinated, organised and recruited participants from the above-mentioned facilities within each municipality. We aimed to take a pragmatic approach and estimated to include between 4–7 participants from each municipality in terms of our resources and to achieve data saturation. As we depended on our contact person for recruitment, we had limited influence on the distribution of participants within different professions. Three pharmacists were included, two of them employed as a community pharmacist, and the third had previous experience as a community pharmacist. All participants received information about the study aims, funding and roles before the interview. We included pharmacists (n = 3), GPs (n = 6), home care nurses (n = 8), nurses (n = 9) and MDs (n = 6) working in nursing homes, a total of 32 participants (Table 2). Initially, 35 participants agreed to be interviewed, although three declined on short notice (nurse n = 2, pharmacist n = 1). 

### 2.3. Data Generation and Analysis

The authors (USM, TTK) conducted the interviews with the participants face-to-face at their workplace during the period of November 2019 to March 2020. Only the authors and the participant(s) were present during the interviews. Most participants were interviewed individually, with the exception of six, which we interviewed jointly in pairs for practical reasons. The interviews lasted between 30–60 min, were digitally recorded and further transcribed by a professional transcribing company. Written informed consent was required and collected from all included participants. 

The analysis and data coding was performed by both authors. We followed a stepwise-deductive-inductive (SDI) approach [29,30] oriented towards identifying emergent issues and themes through an open inductive reading of the material. All transcripts were first coded in close detail to maintain the content of the material using QSR NVivo 12 software [31]. For this study, we created 131 individual nodes covering access to and the exchange of medication information. In the second step, the nodes were arranged into a coding tree consisting of five recurring themes: ‘fragmentation of information systems’, ‘perceived risk of errors’, ‘excessive time use’, ‘dependency on others’ and ‘uncertainty’. In order to ensure consistency and continuity between the empirical data, our open inductive reading, categorisation and grouping, the researchers maintained a constant focus and dialogue to ensure that all of the nodes and themes created both faithfully represented what was actually being reported by the participants in the material and were strong enough to cover all of the included nodes [30,32]. 

Our study was not based on any specific pre-existing theory or set of concepts, which is in line with the logic of Grounded Theory [26,27,33]. The categories, themes and concepts were based on patterns that emerged from the empirical data. Through abductive reasoning [30], an overarching theme was identified that was common to all the original themes. Common to all of these themes was that they, in some way, referred to a state of being cut off from information and that they implied that having access to this information would be preferable.

### 2.4. Ethical Considerations

The study has been approved by the local data protection officer at the University Hospital of North Norway. All participants were recruited based on voluntary participation and signed a written consent form prior to the interviews. The consent form explicitly stated the right to withdraw at any time without providing a reason. The data material was anonymised and handled securely according to the recommendations of the local data protection officer. Participant quotations presented in the results were only identified by profession and type of institution to preserve anonymity. The Consolidated Criteria for Reporting Qualitative Research (COREQ) checklist was used to fulfil the standards of high-quality research (Appendix A). 

## 3. Results

Health professionals in primary health care generally experienced the task of obtaining and sharing information on patient medicine lists as complex and as a significant challenge. The main pattern that emerged from our analysis was that they regularly experienced situations in which access to an accurate and updated medicine list was obstructed and it required extra effort and workarounds to ensure medication safety. Consequently, the state of being cut off from information emerged as a common theme that was present in all of our material relating to the procurement and sharing of medication information. In analysing the work practices of the different health professionals, the following five main themes emerged from the empirical data: Fragmentation of information systemsPerceived risk of errorsExcessive time useDependency on othersUncertainty.

In the following section, we present these themes in more detail.

### 3.1. Fragmentation of Information Systems

A significant number of the challenges expressed by our participants were related to the fragmentation of information caused by digital systems (i.e., EHRs) that were not linked across health organisations and levels of care. This was expressed by a GP as *“systems that do not communicate”*. The challenges were perceived as being cut off from information, causing a disruption of the information flow in communication across health care services. As one GP said, 


*The main problem [with medication information] lies in the transition of care, whether it is from the hospital with a medication list on paper only, information to or from the hospital in the discharge summary or between the home care services and us.*


Some of the MDs working at nursing homes experienced being cut off from information that the GPs and hospital doctors are able to obtain about patient medication. The MDs depend on information actually being shared by both GPs and hospital doctors. As one MD explained, “We are lost in a kind of information shadow between the GPs and the hospital. I don’t think they realise how little information we really have.” Several of our participants used the phrase ‘detective work’ to describe how they took on extra responsibilities and tasks in order to handle the fragmentation of information. A community pharmacist stated, 


*Yes, there is quite a bit of detective work needed to make sure the medication is correct,” while a home care nurse used the phrase to explain how she managed without adequate access to necessary information as “we almost need to be like detectives to get the [medicine] information.*


### 3.2. Perceived Risk of Errors

The fragmentation of digital systems and being cut off from important information was perceived as a risk of medication error and threatening patient safety in different ways across our participants. One GP stated that “there are many stages of the [medication management] process and significant opportunities for error. A [digital shared medication] list that is clear and concise for everyone will save lives.” Most of the nurses expressed that deviations and errors in the medication management process occurred and were quite common, but few had experienced that medication errors actually caused patient harm. One home care nurse stated the following: “There are many errors [in the medication list], but that’s not a problem because it does not pose a threat to the patient’s life.” Another nurse from an intermediate department expressed the following: 


*There have been deviations and patients have not received the medicines they are supposed to have. But, in the end, everything turned out all right, so I have not experienced any life-threatening situations [for the patients]. But, of course, this can be problematic for treatment.*


A common challenge for our participants was the existence of multiple versions of a patient’s medication list, which often differed to some extent between the GP, patient, pharmacy, home care service and nursing home. The community pharmacists also stated that a lack of access to the digital systems (i.e., EHRs and SCR) for information on patient medication lists is a challenge, especially when performing medication revisions in the primary health care services. One of the pharmacists summarised the main challenges as follows: 


*The challenge is the existence of multiple lists and often poor discharge summaries from the hospitals. The [information] flow between different levels of care is the main challenge. And, of course, another challenge is that we (as pharmacists) lack access to this information.*


### 3.3. Excessive Time Use 

Common for most of our participants was the experience of a very time-consuming process for obtaining the necessary medication information in their everyday work. The lack of adequate digital systems for sharing patients´ medicine lists across health care services was emphasised as one of the main reasons for these challenges, as this led to a number of manual tasks being needed to ensure that the information was correct and updated. 

All of the nurses reported that they used an excessive amount of time to obtain medication information, although home care nurses and nurses working in intermediate/short-time departments, in particular, highlighted this as one of the main challenges. This process was especially time-consuming when a patient returned from a hospital stay. One home care nurse explained this process as follows:


*We contact the GP by phone or e-link and ask for the patient’s medication list to ensure that we have the correct information. If there have been any changes, the (medication) list is sent back and forth several times between us and the GP for medication reconciliation.*


All our participants needed to use different digital and manual sources to obtain information about the patients´ medication list. A GP said, “We obtain the information from the patient, discharge summaries and the SCR, so we need to continuously update the medication list.” Other GPs explained that it could be difficult and time-consuming to ensure the accuracy of the medication list, but that they have good routines for this. One GP stated the following: 


*We need to clarify and update the medication lists continuously and this also requires manual work. Most of the time, we are able to get this done, but you need time for this, and you might have 30 discharge summaries a day that you need to go through.*


### 3.4. Dependency on Others

Pharmacists, nurses and MDs highlighted the dependency on other actors (GPs, hospital doctors, the patient or caregivers) to confirm or provide access to correct medication information before they could perform their tasks. When patients were transferred from the hospital to a nursing home, a common challenge was the lack of an approved, or delayed, discharge summary. One MD explained this process as follows:


*I have to call a hospital doctor [...] and this person may not be at work. And when I send a message to the patient’s GP, they are not able to answer me quickly or I have to wait [..]. I spend a lot of time on this that I could have spent on other things, such as working with or examining patients.*


The pharmacists described several tasks in which dependency on others was time-consuming, especially related to obtaining correct medication information or a complete medicine list from GPs. The pharmacy does not have a digital communication system with the GPs’ offices, so they need to either call or fax in most cases when there are questions about a prescription, changes to a patient’s medication list, a signature is missing, or the GP has made corrections to the prescribed medications. This was especially challenging with regard to multidose prescriptions: 


*Sometimes, we receive a paper copy/fax of the medication list and we can see that the GP has not changed a medication from ‘if needed’ to ‘regular use’. It is then our responsibility to call the GP and, sometimes, we may talk to the GP, but often we need to leave a message with the secretary.*


### 3.5. Uncertainty

Many of our participants reported that being cut off from medication information made them feel uncertain. One of the home care nurses described the experience as “receiving the discharge summary printed on paper feels unnecessary and unsafe.” In the absence of direct access to updated information, this participant had to rely on a potentially outdated printout of the information, while also running the risk of physically losing the printed list. Another IMD nurse focused on how the existence of multiple medication lists [for the same patient] created uncertainty, especially when it affected the task of preparing and dosing medications:


*You suddenly realise, “Oh, this is the wrong dosage!” or maybe it has been discontinued. So, when there is a mismatch between the different lists available to you, it creates uncertainty, especially when you are the one [responsible for] putting the pills in the dosette box.*


As we see, dealing with multiple and different medication lists especially made the nurses uncertain about whether they actually had access to the correct information. In other words, the experience of being cut off from obtaining the correct information created an experience of uncertainty. Below, we present examples in which uncertainty is followed by negative emotions and a sense of changed responsibility. When we asked about their experiences of uncertainty, some of our participants reported that uncertainty also made them feel stressed and frustrated and that this affected patients:


*It is time-consuming, so you get frustrated often. And the patients wonder what is going on, they become confused, putting you in a difficult position and forcing you to go hunting for information. (Home care nurse)*


Some nurses also reported a perceived feeling of increased responsibility as a result of the uncertainty that was created by being cut off from information. As one IMD nurse stated: “[Different drug lists] create frustration and a fundamental uncertainty and this leaves you feeling very responsible. After all, you are the one administering these drugs and you actually cannot be sure whether what you´re doing is one hundred percent [correct].” 

Several of the nurses said that when the list in the discharge summary from the hospital differed greatly from the local list in the electronic patient journal, they had to put in a lot of effort to resolve the differences and solve the problem. As a nurse from IMD stated:


*It often requires a lot of work when the drug list from the hospital or the epicrisis is very different from the one in [our system]. We then understand that changes are needed. But there may be other drugs that we think are forgotten or that there is no justification for why they should be discontinued. And it takes a lot of work to consult with the GP, patient and relatives to put together an accurate list.*


Another nurse reported that the constant experience of errors and mistakes in the medication lists and dosing made it clear that she had to be present and take responsibility during several stages of the medication process in order to compensate for the uncertainty: “It only proves the importance of double-checks [of the medications] and that it is not only one [person] who is doing the dosing and that the nurse must be involved in several stages.” 

As presented in the above section, several nurses in our material expressed uncertainty as a result of being cut off from correct information about patients´ medicine use. 

Summarised, our findings show that being cut-off from information about patient medication lists was frequently experienced and emphasised the different types of challenges by all groups of health professionals in our research, albeit to a different extent due to their role in the medication management process. 

## 4. Discussion 

Overall, our results show that health professionals in primary health care regularly experience several challenges in being cut off from information about patient medicine lists. The lack of an adequate digital solution for obtaining and sharing updated medication information was perceived as the main reason for these challenges. Several tasks and workarounds are being performed to ensure that medication lists are as complete, accurate and updated as possible. In the following sections, we discuss our findings in light of relevant previous research.

### 4.1. Fragmentation of Information Systems

Our participants perceived the fragmentation of information and lack of adequate systems for obtaining and sharing medication information as a threat to patient safety. In particular, the risk of causing patient harm due to medication errors that could occur in different stages of the medication management process was highlighted. This is in line with a previous Norwegian study showing that the lack of possibilities for medication information exchange is a threat to patient safety, as it may lead to potentially harmful medical errors [10]. Other studies show the following obstacles for HIE in the clinical work of nurses and MDs: incomplete information, poor medication management and technology issues [8,13,14].

The participating pharmacists in our study highlighted the lack of access to local digital systems when performing medication revisions in primary health care services, and the lack of digital systems for communication between the pharmacies and the GPs offices. In addition, pharmacists working at a pharmacy often need to communicate with other health professionals (doctors and nurses), mostly using manual systems (phone and fax), making the medication management process fragile, inefficient and a threat to medication safety. Fragmentation also led to some of our participants taking the role of ‘information detectives’, working around the system to obtain the needed information, indicating an increased work-load and the incidence of potentially risky workarounds, based on collective research [12].

### 4.2. Perceived Risk of Errors

A disrupted information flow across health care services, often involving multiple versions of patient medication lists, was highlighted as critical and vulnerable, especially when patients were transferred between different levels of care. Incomplete or not yet approved discharge summaries from hospitals were seen as a source of potential errors due to differences in the previous and new medication lists and a time-consuming process for doctors and nurses to update and ensure that the lists were correct. This is in line with a study by Samal et al., who identified the completion of structured discharge summaries as one of the main targets for quality improvement [34]. 

Our participants perceived the risk of causing patient harm due to medication errors as high in the context of current digital and non-digital systems for medication information. Other studies within the primary health care services support our findings, stating that the risk of errors is high in the lack of digital systems able to share medication information [6,12,35]. As such, this highlights the importance of providing all health professionals involved in the medication management process access to relevant, accurate and timely digital systems for medication information to provide high-quality and safe health care.

A Norwegian study investigated errors in current paper-based medication lists, reporting that 88% had discrepancies in the medication list between the GP, home care services and dispensing pharmacy [18]. Furthermore, the authors reported that GPs experienced the medicines reconciliation process and creation of a shared list to be highly time-consuming and the home care service and pharmacy reported numerous errors in the first list created. Another study investigated the discrepancies between the medication list in the EHR and the list in the national prescription repository, showing over 80% for at least one type of discrepancy including: noncurrent, a duplicate or missing prescription [17].

### 4.3. Time Consumption and Dependency on Others

The process of obtaining an accurate medicine list was time-consuming and resource-demanding, reducing the efficiency of patient-related work for all of our participants. The medication management process required several manual tasks and workarounds to ensure that patients received the correct medications. Other international studies have reported obstacles in information exchange within health care services that support our findings, with consequences reported being: an inefficient workflow, incomplete information and shared information not meeting users’ needs [8,13]. A Swedish study on the physician’s view of the transition from a local to regional shared medication list reported that the regional list improved accessibility and accuracy of the information and was less time-consuming, but could decrease the confidentiality of information [36]. The study shows the potential for a shared digital medication list to be less time-consuming compared to the current local lists. 

Excessive time consumption caused by a dependency on others for information was especially prominent for those working in home-care services and intermediate departments at nursing homes. This may be explained by the fact that nurses in home-care and intermediate departments normally do not have an MD present and need to contact other parties (the emergency room, hospital doctors, home care nurses, relatives or GPs) to obtain information. A study investigating the number of medicine-related problems (MRP), comparing home-care services and nursing homes, reported a significantly higher number of MRP in patients receiving home-care compared to patients in nursing homes [15]. This may be due to the fact that home-care nurses are more dependent on several other parties for obtaining the correct information on patients’ medications than nurses in nursing homes. In line with our results, Devik et al. (2018) highlighted the need for future research to explore how different care settings may influence medication safety [15]. 

### 4.4. Uncertainty

Our participants reported that errors, discrepancies and the existence of multiple medication lists—in short being cut off from correct information—created major uncertainties in their everyday work. Identifying the link between access to correct information and uncertainty, we complemented the findings of earlier research showing that errors in discharge summaries from hospitals created uncertainty for both patients and health care workers in home-care [12]. 

The state of being cut off from correct medication information made our participants stressed and frustrated. These findings support earlier studies, which found that health professionals reacted to the difficulties in obtaining correct medication information with frustration and stress [10,12]. An increase in stress and frustration may also have implications for well-being and performance, affecting both health professionals and patients. The experience of uncertainty affected our participants’ perceptions of responsibility, as many of our participants reported taking on new responsibilities in order to be certain that they possessed the information they needed to deliver safe health care. The link between uncertainty and taking on new responsibilities has implications for both the individual workload and the use of resources on the organizational level and supports earlier research by Remen and Grimsmo on how insufficient information causes stress, additional workloads and risky workarounds [10]. 

### 4.5. Overall Distribution of Negative Effects Being Cut off from Information

The experience of being cut off from information, as discussed above, was shared by all professionals in our study. However, some of the problems seemed to be particularly widespread among nurses. Nearly all nurses in our study expressed significant distress over fragmentation, time consumption, complexity, uncertainty, stress, frustration and unwanted extra responsibilities. This finding supports Berlands´ point that parts of the nurses´ problems are attributable to GPs lacking the routines for transferring information to healthcare workers [12]. It also supports a study claiming that hospital routines and professional culture are important factors to consider promoting more transparent health care for patients and improved interdisciplinary communication [37]. Our findings can also be interpreted in the light of the nurses´ position at the bottom of what we may call “the information hierarchy”, at the mercy of other health professionals—typically GPs—higher up in the hierarchy failing to provide them with consistent information. This interpretation conforms with findings from organizational sociology, stating that stress, limitations, lack of flexibility and other constraints impact more severely the lower one´s position in the professional hierarchy [38]. 

For pharmacists, the lack of access to local digital systems when performing medication revisions and in communication with the GPs offices, was the main challenge and perceived as posing a threat to medication safety. Taking into account the important role of pharmacists in the medication management process in primary health care, it seems important to increase the presence of community pharmacists in Norwegian municipalities, and thereby, access to the SML is deemed necessary. This is supported by two Norwegian studies reporting the importance and potential of pharmacists involved in an interdisciplinary team in nursing homes to reduce MRP [39,40]. According to the planned SML, it is still not clear whether pharmacists (working as community pharmacists or at a pharmacy) will get access to the SML.

### 4.6. Consideration of Methodology and Design

This study is in line with the Norwegian eHealth strategies that view research-based knowledge as a key tool in the planning, implementation and evaluation of national initiatives. [25]. Research results from the intersection between health, technology, organisation and society will be used as the basis for decision-making and to shed light on the effect of eHealth interventions [25]. A qualitative research design was used to obtain in-depth information on the views and experiences of health professionals in obtaining and sharing information regarding patient medication lists. Qualitative research is usually conducted to study a problem that has not yet been clearly defined and does not aim to provide the final and conclusive answers to the research questions, but merely explores the research topic with varying levels of depth [41,42]. 

Our study is limited, as the number of participants is relatively small and only six municipalities are included. Only three pharmacists were included, due to our pragmatic approach to recruitment through a contact person within the municipalities. As such, we are not able to describe how widespread the identified perceptions are among health professionals in Norway. However, the identified challenges presented in this article may be similar and transferrable to other countries planning to implement a national shared medication list, especially in countries with a similar health and social care system as Norway. The six included municipalities with 32 participants may serve as a cross-section of the Norwegian primary care services, as we included participants from municipalities with a geographical spread, different sizes of municipalities, different professions using different eHealth systems. To safeguard the privacy of participants, we were not directly involved in the recruitment process, which was handled by our contact persons in each municipality. We chose to contact each municipality through its health care administrators in order to establish a good and solid relationship with the municipalities, to comply with research ethic rules and avoid obtaining personal information about the participants. All in all, we have assessed the risk of participation bias to be low in this study. 

The further steps in this pre-study are to investigate the experiences of a larger population of health professionals (survey) in order to acquire a broader knowledge base on experiences with access to medication information in current digital and non-digital solutions. The upcoming survey is developed based on the results of our interviews in this pre-study. In addition, patient perspectives through interviews and surveys will be conducted during 2021. The pre-study serves as a baseline for future research on the effects during and after the implementation of the SML in primary health care. This is important in order to identify problems that may disrupt the implementation, workflow, communication and information flow between health care organizations and levels of care before a full national rollout. This approach is supported by a recent review of factors for the success and failure of eHealth interventions; there is a critical need to perform in-depth studies of the workflow(s) that an intervention is supposed to support and to examine the clinical processes involved [43]. 

## 5. Conclusions

This present study provides important insight into the experiences of nurses, pharmacists and doctors with the work process of obtaining medication information. Our study found several challenges caused by being cut off from the necessary information, posing a threat to patient safety and the quality of care delivered. The current digital and non-digital systems for obtaining medication lists were perceived as fragmented, complex, risky and time-consuming, as well as causing uncertainty. The challenges were especially related to the critical phase of patient transition between levels of care. 

On the basis of our findings, we hope to contribute to a sense of urgency among decision-makers about the present situation and the need for change. Health professionals in Norwegian primary health care are in critical need of a new shared digital medication list in order to prevent medication errors. Our results are important in order to monitor the effects, limitations and possibilities of ongoing and planned national eHealth initiatives. To achieve the health political goals of improved quality, increased patient safety and efficiency, there is a need to study the opportunities offered by eHealth solutions before a national rollout. 

## Figures and Tables

**Table 1 pharmacy-09-00046-t001:** Excerpt from interview guide.

Theme 2: Access to Medication Information
How do you proceed to obtain an overview of which medications a patient is using?
How do you obtain and communicate changes to a patient’s medication list?
What are the current challenges in relation to the lack of a possibility to share medication lists?

**Table 2 pharmacy-09-00046-t002:** Description of included municipalities and participants.

No.	Size of Municipality	No. of Citizens	Pharmacist	General Practitioner	Home Care	Nursing Home	Doctor, Nursing Home
**1**	Medium	20,500	1	0	2	2	1
**2**	Large	64,000	1	1	2	3	0
**3**	Medium	30,000	0	2	2	2	1
**4**	Large	50,000	1	1	0	1	2
**5**	Small	5000	0	0	1	1	1
**6**	Medium	13,000	0	2	1	0	1

## Data Availability

The data generated and analysed during the current study is not publicly available due to privacy reasons, but available from the corresponding author on reasonable request.

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
