# Peer review of "Challenges Faced by Health Professionals in Obtaining Correct Medication Information in the Absence of a Shared Digital Medication List"

_pharmacy, 2021, doi:10.3390/pharmacy9010046_

Round 1

Reviewer 1 Report

This study seeks to investigate the experiences of health professionals and patients with the medication management process before the implementation of the Shared Medication List (SML) and I find the topic very relevant as this is the problem faced by many (if not all European) countries.

Participants

You claimed to take a pragmatic approach and estimated to include between 4-7 participants from each municipality; you included pharmacists (n=3) , GPs 141 (n=6), home care nurses (n=8), nurses (n=9) and MDs (n=6) working in nursing homes, a total of 32 participants. Can you please justify such a distribution of participants/health care professionals? Based on which pre-specified criteria did you settle with 3 pharmacists as opposed to 17 nurses? Did you expect different results based on representation of different health care professions?

For pharmacists, the lack of access to local digital systems when performing medication revisions and in communication with the GPs offices, was a main challenge and perceived as posing a threat to medication safety. However, than you claimed that according to the planned SML, it is still not clear whether pharmacists (working as community pharmacists or at a pharmacy) will get access to the SML.

How do you comment on that? It seems that regardless of your results, the plan on dealing with this problem (getting digital solutions to ensure seamless, up-to-date information about patient medication lists) is already pre-determined and the results obtained from this study will not significantly influence that.

Grammar

Please review the grammar in this sentence: 

Of the seven included municipalities, planned interviews for the pre-study was not possible for one due to unexpected circumstances.

Author Response

You claimed to take a pragmatic approach and estimated to include between 4-7 participants from each municipality; you included pharmacists (n=3) , GPs 141 (n=6), home care nurses (n=8), nurses (n=9) and MDs (n=6) working in nursing homes, a total of 32 participants. Can you please justify such a distribution of participants/health care professionals? .

Based on which pre-specified criteria did you settle with 3 pharmacists as opposed to 17 nurses?

This is now explained in line 140-143 and line 491-492.

Did you expect different results based on representation of different health care professions?

No. Our goal was to represent different health professionals working with medication management in primary health care and to capture their experiences.

For pharmacists, the lack of access to local digital systems when performing medication revisions and in communication with the GPs offices, was a main challenge and perceived as posing a threat to medication safety. However, than you claimed that according to the planned SML, it is still not clear whether pharmacists (working as community pharmacists or at a pharmacy) will get access to the SML.

How do you comment on that? It seems that regardless of your results, the plan on dealing with this problem (getting digital solutions to ensure seamless, up-to-date information about patient medication lists) is already pre-determined and the results obtained from this study will not significantly influence that.

It is a fact that we don’t know if pharmacist will get access to the new digital solutions, as this is not decided yet. Therefore, our findings might be of importance to the authorities and influence the decisions of whether the pharmacists will be able to access the PLL.. See line 471-474 in our manuscript.

Grammar

Please review the grammar in this sentence: 

Of the seven included municipalities, planned interviews for the pre-study was not possible for one due to unexpected circumstances.

See line 123, changed to were

Reviewer 2 Report

This is a very interesting article especially in light of the fact that it is part of a much larger overarching study with regard to a shared national medication list and thus will inform and ideally contribute to informed policy  In that context i feel the paper would have additional robustness by reference to the WHO Global Health Challenge  to reduce avoidable medication related harm by 50% over the next 5 years to which most countries have signed up.  A key technical document within this aim relates to transitions of care  and the issues and challenges that occur  Thus the inclusion of reference to this and some pertinent statistics and associated references   in the introduction  would add weight to the concept of this matter as a  policy imperative as well as broadening the geography of the problem per se.  Thus this work could be to some degree generalisable and transferrable to other healthcare systems.

Author Response

We have added your suggestions of new references and information in line 32-34.